# Mitochondrial Control of Genomic Instability in Cancer

**DOI:** 10.3390/cancers13081914

**Published:** 2021-04-15

**Authors:** Massimo Bonora, Sonia Missiroli, Mariasole Perrone, Francesco Fiorica, Paolo Pinton, Carlotta Giorgi

**Affiliations:** 1Section of Experimental Medicine and Laboratory for Technologies of Advanced Therapies (LTTA), Department of Medical Sciences, University of Ferrara, 44121 Ferrara, Italy; bnrmsm1@unife.it (M.B.); msssno@unife.it (S.M.); prrmsl@unife.it (M.P.); pnp@unife.it (P.P.); 2Department of Radiation Oncology and Nuclear Medicine, AULSS 9 Scaligera, 37100 Verona, Italy; francesco.fiorica@aulss9.veneto.it

**Keywords:** mitochondria, genomic instability, tumor progression, mitophagy, p53, ROS, calcium, apoptosis

## Abstract

**Simple Summary:**

Cancer cells display among its hallmark genomic instability. This is a progressive tendency in accumulate genome alteration which contributes to the damage of genes regulating cell division and tumor suppression. Genomic instability favors the appearance of survival-promoting mutations, increasing the likelihood that those mutations will propagate into daughter cells and have a significant impact on cancer progression. Among the many factor influencing this phenomenon, mitochondrial physiology is emerging. Mitochondria are bound to genomic instability by responding to DNA alteration to trigger cell death programs and as a source for DNA damage. Mitochondrial alterations prototypical of cancer can desensitize the mitochondrial route of cell death, facilitating the survival of cell acquiring new mutations, or can stimulate mitochondrial mediated DNA damage, boosting the mutation rate and genomic instability itself.

**Abstract:**

Mitochondria are well known to participate in multiple aspects of tumor formation and progression. They indeed can alter the susceptibility of cells to engage regulated cell death, regulate pro-survival signal transduction pathways and confer metabolic plasticity that adapts to specific tumor cell demands. Interestingly, a relatively poorly explored aspect of mitochondria in neoplastic disease is their contribution to the characteristic genomic instability that underlies the evolution of the disease. In this review, we summarize the known mechanisms by which mitochondrial alterations in cancer tolerate and support the accumulation of DNA mutations which leads to genomic instability. We describe recent studies elucidating mitochondrial responses to DNA damage as well as the direct contribution of mitochondria to favor the accumulation of DNA alterations.

## 1. Introduction

Mitochondria are highly dynamic organelles that are quintessential for eukaryotic cells since they mediate fundamental activities indispensable to cells’ health. As the center of oxidative metabolism, mitochondria are the powerhouse of the cells, they participate in calcium (Ca^2+^) homeostasis, they are the principal source of reactive oxygen species (ROS) production and are involved in regulated cell deaths (RCD) [1,2]. Additionally, mitochondrial functions are fine-tuned by a delicate balance between mitochondrial dynamics (fusion and fission), biogenesis and autophagy. Moreover, mitochondria are a core component of the signal transduction cascade, since they act as a signaling platform that receives cellular signals and propagates targeted responses.

Reflecting the key role of mitochondria in all these balanced and perfectly organized activities, mitochondrial perturbations, modifications of their key components or defects in this complex machinery are correlated to several diseases [3,4,5]. Cardiovascular diseases, neurological disorders and metabolic abnormalities are just a few examples of the multitude of pathologies in which mitochondria are involved. Last but not least, mitochondrial alterations are closely related to cancer [6,7].

Mitochondria harbor their own circular and double-stranded DNA genome (mtDNA), independently from the nuclear one (nDNA), which encodes 37 genes: 22 tRNAs, 2 rRNAs and 13 proteins necessary for electron transport chain (ETC) and oxidative phosphorylation (OXPHOS). Mutations in mtDNA have been found in a spectrum of human cancers, such as hepatocellular carcinoma, breast and gastric cancers and colorectal tumors, correlated with defects in mitochondrial respiration, high lactate levels and increased tumorigenesis [8,9,10,11].

Structurally, mitochondria are double-membrane bound, organized as an outer membrane (OMM) permeable to ions and metabolites up to 5000 Da, an intermembrane space (IMS) and a highly selective inner membrane (IMM), characterized by invaginations called cristae, which enclose the mitochondrial matrix and where mitochondrial ETC complexes (as well as other proteins) are hosted and protected from random diffusion [12]. The inner boundary membrane is enriched with structural proteins and components of the import machinery of mitochondria.

Functionally, mitochondria continuously undergo fission, regulated mainly by dynamin-related protein 1 (DRP1), or fusion, mediated by mitofusins 1 and 2 (MFN1 and MFN2) and optic atrophy 1 protein (OPA1), in a balanced manner to regulate their overall morphology and to adapt to their energetic needs [13]. Interestingly, OMM proteins have vital roles in monitoring mitochondrial quality and are targets of a specialized autophagic pathway (mitophagy) that ensures the selective removal of dysfunctional or damaged mitochondria [14,15]. Since mitochondrial biology and metabolic plasticity have a central role in maintaining cellular homeostasis and physiological demands, it is not surprising that cancer cells modify these mitochondrial dynamics to support the high bioenergetic demand and to improve their proliferation and survival.

As already mentioned, mitochondria occupy a central position in the control of RCD, where mitochondrial outer membrane permeabilization (MOMP), driven by the activation of proapoptotic effectors of the B cell lymphoma 2 (BCL2) family of proteins (notably BAX and BAK) initiates a cascade that leads to the opening of pores on the OMM and release of the pro-apoptotic factor’s cytochrome c, endonuclease G (Endo G), apoptosis inducing factor (AIF), Smac/DIABLO. These proteins, once in the cytoplasm, can induce apoptosome formation, caspase and nucleases activation leading to RCD execution [16,17,18]. Apoptosis is not the only mechanism for RCD; other types are emerging such as necroptosis, pyroptosis and ferroptosis, which include pro-inflammatory signaling induction [19,20,21].

Mitochondria are structurally and functionally connected to the endoplasmic reticulum (ER, the major cellular store for Ca^2+^) through the mitochondria associated membranes (MAMs), specialized regions that provide a molecular platform that decode different cellular functions between the two organelles [22]. MAMs play a pivotal role in different signaling pathways, including lipids transfer, autophagy, Ca^2+^ homeostasis, apoptosis and inflammatory responses [23,24,25]. Alterations in the composition of MAMs lead to several pathological conditions, including cancer [26,27,28].

Furthermore, mitochondria are the primary source of reactive oxygen species (ROS) through respiratory complexes (mainly complex I and III), but also via NADPH-oxidase, monoaminoxidase, aconitase and others [29,30]. In normal cells, ROS participate in stress signaling; however, in cancer cells mitochondrial ROS induce nDNA and mtDNA damage and activate cancer-promoting transcription factors, thereby promoting neoplastic transformation [31,32].

The interconnections between these mitochondrial functions and diseases, including cancer, have been widely described and studied; however, little is known about mitochondrial dysfunctions and genomic instability. This area of study is still in its early stages, with several studies starting to explore and pinpoint the importance of these interconnections in cancer.

The purpose of this review is to critically discuss how genomic instability and mitochondrial homeostasis are closely related to each other and how this interaction is important for cancer progression, focusing on the possibility of harnessing such defects for therapeutic purposes.

## 2. Mitochondrial Response to nDNA Damage

Cells are continuously subjected to a large variety of DNA lesions, which can be generated endogenously (e.g., by oxidative stress) or exogenously (such as ultraviolet light, ionizing irradiation or DNA damaging drugs). Furthermore, different types of reactive species such as ROS, reactive nitrogen species, alkylating agents and lipid peroxidation produced by metabolic reactions can be harmful for DNA. There are a great variety of DNA damage, ranging from single or two-bases alternations to double or single strand DNA breaks and cross-links between bases in the same or opposite strands or between DNA and protein molecules.

Depending on the type of DNA damage, cells can activate several strategies of contingency. DNA damage leads to cellular responses like cell-cycle arrest, DNA repair, senescence, or apoptosis, which are referred to as DNA damage response (DDR). DDR includes a sophisticated network of signaling pathways that cells could engage in depending on the context in which the damage occurs. The proficient execution of DDR prevents the transmission of harmful mutation and genomic instability to the progeny of the affected cells. Apoptosis is an ultimate cellular decision which consists in the physical loss of the damaged cells that would increase the risk of genomic instability and cancer.

As previously described, mitochondria play a key role in apoptosis execution. Indeed, intrinsic stimuli such as metabolic, replicative, and genotoxic stress result in the induction of the mitochondrial apoptotic pathway.

The most famous responder to DNA damage is p53, whose role in the cell fate can be either pro-death or pro-survival, depending on the level of DNA damage, duration of the DDR and cell type. In general, high levels of DNA damage that results in persistent p53 activation trigger RCD, while low levels activate p53 only transiently, promoting repair and survival mechanisms (Figure 1). DNA damages activate ATM/ATR and CHK1/CHK2 kinases that phosphorylate p53 at the N-terminus, followed by its subsequent stabilization and activation. Among its numerous activities, p53 can move to mitochondria which directly promotes MOMP, inducing both mitophagy and apoptosis, under a variety of RCD-inducing conditions [33]. Notably, many oncogenic mutations affect p53 as well as its mitochondrial partners, causing the failure of apoptosis engagement and promoting tumor formation [34,35,36].

p53 can also suppress the activity of mitophagy executors PINK1 (PTEN-induced kinase 1) and Parkin. In particular, nuclear p53 suppresses the expression of PINK1 [37] while cytoplasmic p53 interacts with Parkin, interfering with its translocation to mitochondria [38], then favoring the accumulation of dysfunctional mitochondria prone to engage MOMP.

Mitophagy can also be positively modulated by DDR pathways (Figure 1). The Fanconi anemia complementation group C (FANCC) protein interacts with mitophagy mediator PARKIN, promoting mitochondrial clearance [39]. Further, mitochondrial dysfunction due to impaired mitophagy has been reported in cells lacking ATM. Multiple mechanisms could be accounted for this phenotype. ATM deletion causes extensive activation of PARP1 which leads to depletion of the NAD+ pool and inhibition of several sirtuins. These are NAD^+^-sensitive promoters of autophagy that act via multiple pathways (for a more comprehensive review see [40]). Furthermore, ATM can sustain mitophagy via the NEMO-JNK pathways [41]. Considering the discrepant observations reported for the master responders of DDR, p53 and ATM, the net effect of DNA damage on mitophagy and its impact in genetic instability and tumor formation is still an open question.

A relatively novel player in DDR and tumor formation is Ca^2+^ signaling. Indeed, cancer-driving oncogene and tumor suppressor mutations exert their pro-tumorigenic functions by altering normal Ca^2+^ homeostasis [42]. Thus, the malignant remodeling of Ca^2+^ dynamics helps to sustain cancer hallmarks [43]. Indeed, many studies have contributed to defining the role of cytoplasmic Ca^2+^ levels in different phases of tumor progression [44]. Some plasma membrane Ca^2+^ transporters that promote Ca^2+^ entry from the extracellular milieu, such as Orai3 channels [45,46], are currently considered pivotal factors in cancer development. The bigger players in organelle Ca^2+^ communication are the endoplasmic reticulum (ER) and mitochondria. Both of them are strictly associated with each other and form membrane tethers important for Ca^2+^ transfer and the exchange of other ions and phospholipids.

The most accepted physiological role of mitochondrial Ca^2+^ uptake is the control of metabolic activity of the mitochondria. Indeed, different members of the TCA cycle are activated by Ca^2+^ via different mechanisms. Those enzymes (α—ketoglutarate dehydrogenase, isocitrate dehydrogenase and pyruvate dehydrogenase phosphatase) represent rate-limiting steps thus controlling the feeding of electrons into the respiratory chain and the generation of the proton gradient across the inner membrane, in turn necessary for Ca^2+^ uptake and ATP production [47].

It has also been demonstrated that Ca^2+^ transfer from mitochondria to ER is causally linked to RCD. Specific circumstances (e.g., hyperstimulation of ionotropic glutamate receptors, high ROS, or ER stress) lead to Ca^2+^ cycling across the mitochondrial membranes, collapse of the proton gradient and bioenergetic catastrophe, thus leading to RCD by necrosis. Undoubtedly, Ca^2+^ binds to subunit beta of mitochondrial F1/FO ATP synthase, favoring its transition to a death channel called mitochondrial permeability transition pore (mPTP) [48,49]. mPTP opening cause the collapse of mitochondrial homeostasis inducing the impairment of intracellular activities, culminating in necrotic phenotype. In addition, mitochondrial Ca^2+^ sensitize cells to apoptotic engagement, mPTP allows the release in the cytosol of intermembrane-residing apoptotic factors triggering caspase-dependent and a caspase-independent apoptosis. The ER-resident Ca^2+^ channel inositol phosphate receptor (Ip3R) has been demonstrated in different cell types to mediate this Ca^2+^-regulated RCD, especially its isoform 3 (Ip3R3). The ER-mitochondria interface, i.e., MAMs, is the residency of Ip3R3 and of several cancer related factor, such as the tumor suppressors PML [50] and PTEN [51] and the proto-oncogene AKT. These can localize to MAMs where exert their pro- or antiapoptotic activities by altering Ip3 dependent Ca^2+^-transfer and RCD. AKT phosphorylate Ip3R3 and the MCU complex component, MICU1, impairing their activities [52,53]. PML and PTEN counteract the Ip3R3 phosphorylation by favoring its interaction with the phosphatase PP2A. Moreover, PTEN can stabilize Ip3R3 competing for its binding with the F-box protein FBXL2, a receptor for Ip3R3 ubiquitination [54]. In addition, p53 can localize at ER where it interacts with SERCA (Sarco-Endoplasmic Reticulum Calcium ATPase)-2, the pump that maintains elevated the Ca^2+^ concentration within ER lumen. The interaction with wild type, but not cancer-related mutant, p53 favors SERCA-2 activity and maintains an effective Ca^2+^ dependent RCD [55,56]. In contrast, Bcl-2 overexpression has been demonstrated to favor Ca^2+^ leak from ER, dampening the execution of apoptosis [57]. Again, it has been proposed that H-RAS impairs tumor transformation by Caveolin-1-dependent modulation of Ca^2+^ within ER [58] (Figure 2).

Recently it has been demonstrated that BRCA1-associated protein 1 (BAP1) binds and stabilizes Ip3R3, modulating Ca^2+^ release from the ER to mitochondria, promoting RCD. Most importantly, ultraviolet radiation induced ER Ca^2+^ release, which was significantly impaired in BAP1^+/−^ cells. As a result, BAP1^+/−^ displayed increased DNA damage and survival which was reversed by Ip3R3 overexpression [59]. While the exact mechanism by which DDR activates Ca^2+^ transfer is still to be elucidated, these results indicate that mitochondrial Ca^2+^ uptake is an active mechanism in the control of DDR induced RCD and that its deactivation is instrumental to tumor formation.

## 3. Mitochondria as Cause of Genomic Instability

Mitochondria represent one of the most important sources for ROS generation. High levels of ROS lead to mTORC1 inactivation which promotes mitophagy, reducing the number of mitochondria and through a feedback mechanism, prevents ROS increment [60].

Mitochondrial ROS has a multifaceted and pleomorphic role in DDR because beyond causing damage, they also activate a stress response (Figure 3). Mitochondria-produced ROS can initiate a transduction signaling that leads to transcriptional changes in the nucleus through a retrograde mechanism. Retrograde signaling can induce stress-defense responses including removal of initial dangerous signaling molecules, like ROS. Interestingly, autophagy could be activated by ROS and prevent oxidative stress [61,62]. In general, these “good side” of ROS, especially the mitochondrial one, can be considered as a protective mechanism that help cells from subsequent larger stresses [63].

ROS can induce DNA damage through oxidizing nucleoside bases, which can lead to G-T or G-A transversions. Oxidized bases are typically recognized and repaired by the Base Excision Repair (BER) pathway, but when they occur simultaneously on opposing strands, attempted BER can lead to the generation of DSBs [64]. ROS oxidize nucleotides affect polymerase activity, interfering with the replication fork [65,66]. ROS can also affect replication fork progression through the dissociation of peroxiredoxin2 oligomers (PRDX2). PRDX2 forms a replisome associated ROS sensor that binds to TIMELESS, a fork accelerator. Elevated ROS has led to dissociation of PRDX2 and TIMELESS, thus slowing replication fork speed [67]. Fork breakdown ultimately can lead to DSBs with concomitant genomic instability.

ROS-induced DNA damages have been largely associated to neoplastic disease. Cancer cells have increased ROS production because of regulation of the ROS scavenging system [68] as well as alterations in key signaling pathways related to cellular metabolism. In addition, ROS production has been demonstrated to be instrumental to cancer development and progression in multiple settings [69,70]. Overexpression of the prototypical oncogene c-Myc was demonstrated to induce metabolic remodeling, ROS production and DNA lesions in normal human fibroblasts. Most interestingly the exposure to the antioxidant n-acetyl cysteine (NAC) was able to limit the accumulation of DNA damage, demonstrating the ROS connection between oncogene activation and genomic alterations [71]. Accordingly, RAS overexpression induces ROS production and DNA damage in human normal fibroblasts, and this phenomenon is inhibited by the administration of the inhibitor of mitochondrial respiratory complex I, metformin [72].

Hürthle cell carcinoma (HCC) is a form of thyroid cancer with a marked increase in mitochondria. HCC exhibits both a failure to concentrate radioactive iodine and avidity for fluorodeoxyglucose, an imaging signature suggestive of metabolic reprogramming. HCC is also characterized by large whole-chromosome instability, resulting in a near-homozygous genome (NHG). An NHG is the result of a near-haplodization process, a phenomenon by which a cell population loses one copy of nearly all chromosomes with the consequent loss of heterozygosity for a significant number of genes. Two recent independent studies analyzed the mutational profile of HCC and observed that mtDNA complex I mutations are enriched in HCC compared with a pan-cancer analysis [73,74] and that these mutations are early clonal events maintained during tumor evolution. Comparably, the NHG state was also among the most frequently observed by phylogenetic analysis, suggesting that it also arises early. Interestingly, alterations in the balanced expression of subunits of the respiratory complex I (a concerted activity between nuclear and mitochondrial genomes) are a driving cause of increased ROS production. Thyroid carcinoma cell lines with marked NHG displayed higher ROS production an increase in chromosome segregation errors compared with cell lines with normal ploidy. Furthermore, the exposure to NAC significantly diminished the proportion of mitotic errors [75]. This observation indicates that NHG might promote genomic instability by favoring the selection of mtDNA complex I mutation. To date, we have no information about why NHG is associated with accumulation of mtDNA mutation. One possibility is that complex I mutations are the earliest event and NHG arises by the increase in ROS production and mitotic errors. Still, many neoplastic lesions with mutations in respiratory complex I mitochondrial genes do not display a near-haploid genome. Another possibility is that the remodeling of cellular metabolism induced by complex I alterations provides an unknown selective advantage for cells approaching the NHG. For example, a defect in respiratory chain also causes alterations in the one-carbon metabolism pathway. This pathway connects mitochondria activity to nucleotide synthesis and DNA repair mechanisms and a defect in respiratory complex I could result in impaired cell proliferation restored by some outputs of the one-carbon metabolism. It could be speculated that NHG could survive complex I alterations because of the smaller amount of total DNA that could limit the dependency on one-carbon metabolism. Further experiments are nonetheless needed to clarify if NHG could unmask a real selective advantage for the metabolic consequences of complex I mutation.

Mitochondrial ROS also have a significant impact on tumor development because of autophagy deregulation. Deregulation of autophagy has been demonstrated in multiple neoplastic diseases because of the mutation or adaptive response and offers different forms of selective advantage (for a more extensive discussion of autophagy in cancer please refers to [76]). Genetic inhibition of autophagy via beclin1 deletion favors ROS production and chromosomal instability in response to metabolic stress in a NAC-dependent fashion [77].

A similar mechanism has been proposed in the mutational evolution characteristics of the transition between pre-leukemic conditions to acute myeloid leukemia (AML). Hematological conditions that significantly increase the risk of secondary AML (sAML) are generally referred to as pre-leukemia. These include overt alterations of normal hematopoiesis (e.g., myelodysplastic syndromes, MDS) but also the clonal expansion of HSC (hematopoietic stem cell), which does not significantly affect blood composition. An early mutational event induces aberrant self-renewal of an HSC which makes clones. Through this event (or thanks to an additional one) the hematopoiesis from the expanded clone would result in the aberrant accumulation of myeloid progenitor cells which have limited differentiation capacity [78]. Mutations that characterize pre-leukemia are also typical of sAML and are mostly affect genes related to different levels in the management of genetic information (transcription factors, splicing factors, or epigenetic remodelers). It has thus been proposed that their mutation alters large informational programs involved in the control of HSC self-renewal and differentiation [79]. The expanded clone can eventually acquire a “driver” mutation that induces the neoplastic phenotype, characterized by altered blood count, high frequency of blasts, and more. [80,81].

The mechanism by which pre-leukemic cells acquire a second mutation is still under investigation, though an autophagy-ROS axis was recently proposed. Significantly, Park and co-workers reported that oncogenic mutants of the splicing factor U2AF35 promote the use of a distal poly(A) site in the mRNA of Atg7, a fundamental component of the autophagic machinery. Cells expressing oncogenic mutation cause the inefficient translation of ATG7, impairing the autophagic process. This induces the accumulation of dysfunctional mitochondria, elevated ROS production and spontaneous mutation frequency. Further, exposure to ascorbate (a molecule with antioxidant properties among many other) significantly suppressed the ability of mutant U2AF35 to promote transformation [82]. The fact that elevated ROS production has been reported for AML as well in pre-leukemic conditions [83] supports the hypothesis that mitochondrial ROS might be involved in AML evolution also in response to mutations different from the one already described.

Mitochondrial stress was proposed as source of genomic instability independently on ROS production (Figure 3). As previously described, in specific conditions, the mitochondrial outer membrane can undergo MOMP, releasing in the cytoplasm proteins that activate the apoptotic machinery. Still, at low doses, some apoptotic stimuli can induce sub/lethal MOMP. This occurs when apoptotic stimuli can trigger MOMP in a limited fraction of the mitochondrial network (indicatively, below 10% of total mitochondria). Sublethal MOMP was demonstrated to occur in transformed cells exposed to the Bcl-2 inhibitor ABT-737 or mild expressions of recombinant tBID. In these conditions, while unable to trigger the apoptotic process, the released proapoptotic factors were still able to induce apoptosome formation and caspase-activated DNase. The latter can cause nuclear DNA damage and genomic instability. Most importantly, immortalized (but not transformed) mouse embryonic fibroblasts surviving the ‘failed apoptosis’ were prone to undergoe transformation and demonstrated a higher tumorigenic potential in xenograft assays [84].

A similar mechanism has been proposed as result of autophagic inhibition. Indeed, it was recently demonstrated that after IR-induced DNA damage, autophagy removes damaged mitochondria, hindering the release of the Endo G. In contrast, in autophagy-deficient cells, the release of Endo G, sustains the accumulation of DNA damage and increases genomic instability in cells escaping apoptosis [85]. Whether this mechanism is concurrent or alternative to ROS-induced DNA damage remains to be clarified.

Common fragile sites (CFSs) are large chromosomal regions that exhibit breakage on metaphase chromosomes upon replication stress. They become preferentially unstable at the early stage of cancer development and are hotspots for chromosomal rearrangements in cancers. They are often associated with deletions of tumor suppressor genes and amplification of oncogenes [86,87], and are highly prone to the occurrence of copy number variation [88]. The transcription of genes at CFSs (especially of large one) can interfere with DNA replication, modifying the dynamics or promoting the formations of secondary structures which ultimately lead to fork stalling and incomplete replication [87]. Gene transcription at CFS is also used as readout of CFS instability.

Among DNA replication and repair proteins, members of the FANC pathway (encoded by the FANC genes) function as master regulators of CFS maintenance. Inhibition of mitochondrial respiration by sodium azide or low oxygen exposure, represses CFS expression in basal conditions or after induction by genetic inactivation of FANCD2. On the contrary, causing mitochondrial stress and exacerbating respiration by exposure to the mitochondrial uncoupler CCCP increases spontaneous CFS expression. CCCP impairs homeostatic mechanisms of mitochondrial transport and induces mitochondrial unfolded protein response (UPR^mt^). Genetic inactivation of the UPR^mt^ mediators ATF-4 and UBL5 impairs the CFS expression induced by FANCD2 silencing. This evidence indicates that UPR^mt^ participates to CFS expression and that FACND2 opposes this mechanism. As high CFS translation increases the risk of CFS breakage, this mechanism indicates that UPR^mt^ connect mitochondrial stress to genomic instability induced by replication stress at preferential sites [89]. In agreement, elevated UPR^mt^ response was recently associated with higher tumor aggressiveness and poor patient survival [19].

Ultimately, mitochondrial control of genomic stability might pass through regulation of the epigenome. The epigenetic code is the result of histone modifications and DNA methylation which have a significant impact on gene expression as well as proper organization and management of chromosomes. A number of studies have reported that neoplastic lesions are, non-surprisingly, characterized by large variations of the epigenomic signature. While the impact of epigenetic alterations in tumorigenesis via altered gene expression is an easy mechanism to propose and demonstrate, its connection with genomic instability is a bit more complex and multiple mechanisms could be involved. Indeed, both histone modification and DNA methylation can affect (i) chromosome condensation and centrosomes alignment (therefore proper segregation in mitosis), (ii) stability of the DNA structure, therefore sensitivity to conditions which predispose to breakage, (iii) the mechanisms of DNA repairs, by regulating the activity and recruitment of DNA repairs systems on the lesion or by regulating the expression of DNA repair genes and (iv) controlling telomers length and stability.

Histone deacetylase inhibitors or knockdown/knockout of epigenome modifiers can induce aneuploidy, chromosomal translocations, and copy number alterations in mouse tumor models and human cancer cell lines [90,91,92]. Further DNA hypomethylation and genomic alterations are associated in human cancer [93,94,95,96,97] and global demethylation in the repetitive regions of the genome, were reported to occur early during tumorigenesis, predisposing cells to genomic instability [98].

Interestingly, mitochondria have been largely related to the control of the epigenome. In particular, mitochondria can (i) provide citrate and/or Acetyl-CoA for the activity of histone acetylase; (ii) regulate cytoplasmic α-ketoglutarate therefore the activity of histone and DNA demethylase sensitive to this metabolite, especially Jumonji C domain-containing (JMJD) and Ten-eleven Translocation (TET); (iii) finally, S-adenosyl methionine is the source of methyl groups used by histone and DNA methyltransferases (HMTs and DNMTs), respectively.

In several cancers, mitochondrial alterations have been linked to alteration to the epigenetic landscape [99]. While these alterations could be attributed to all the mechanisms listed above, most evidence indicates that the metabolic rewiring occurring in neoplasms induces the accumulation of fumarate and succinate or of the onco-metabolite 2-oxy-glutharate that inhibits the function of histone and DNA demethylase [100,101].

Because of all these observations, the speculation it could appear as a simple Aristotelian logic that mitochondrial alterations could induce in genomic instability by the generation of a peculiar epigenetic landscape. Still, direct evidence of this mechanism (or its eventual targetability) is lacking.

## 4. Conclusions

In this review, we discussed the major connections between mitochondrial activity and nuclear DNA stability. These can be summarized as follows: (1) DDR can signal to mitochondria, mainly to promote the execution of RCD and (2) mitochondrial stress can directly or indirectly induce nuclear DNA alterations. In addition, we enlightened how multiple mitochondrial alterations observed in neoplastic cells outline in the alterations of these mechanisms. In particular, cancer cells select impaired mitochondrial engagement of RCD in response to DNA damage and exacerbate mitochondrial phenotypes that can potentiate DNA damages (Figure 4). These concepts have a profound meaning for the mechanism of tumor initiation. Indeed, mitochondria not only can become permissive to the propagation of DNA lesions to the progenies but also can support a positive feedback that maximizes the occurrence of large genomic alterations and, ultimately, the establishment of the genomic instability which characterizes most malignant lesions.

The contribution of mitochondria to nuclear DNA stability might have significant clinical implications. Knowing what mitochondrial alterations are linked to genome instability might help to provide better diagnoses of neoplastic disease. In the first place, it could help in predicting the rate of acquisition of new mutations, that in many human malignancies correlates with disease aggressiveness. In the second place, it might inform on the decision of a therapeutic regimen. Indeed, the use of DNA damaging agents is still routinely used worldwide for treatment of most cancers; the preliminary evaluation of mitochondrial status in the lesion could discourage the use of these agents in favor of more modern approaches. Finally, novel therapeutic strategies that revert mitochondrial alterations linked to genome instability could be designed to support conventional chemotherapy or radiation therapy. Indeed, emerging studies suggest that some drugs that target mitochondrial pathways could improve radiosensitivity [102]. For instance, it has been demonstrated that metformin reduced lung cancer cell growth and sensitized them to radiation by modulating the ATM-AMPK signaling [103]. Moreover, it has also been reported that metformin sensitized p53-deficient colorectal cancer cells to radiation by repressing the expression of DNA repair systems and accumulating cells in G2/M phase [104]. In addition, dichloroacetate, a pyruvate dehydrogenase kinase (PDK) inhibitor, was able to alter glioblastoma cell metabolism, activating oxidative phosphorylation and reversing the radiotherapy-induced glycolytic shift [105]. Furthermore, it has been showed that pyrazinib (a pyrazine phenol small molecule drug with anti-angiogenic and anti-metabolic activity) stimulated radiosensitivity in a model of radioresistant esophageal adenocarcinoma by modulating mitochondrial bioenergetics [106].

In conclusion, while this field of study still requires extensive study for complete molecular characterization, understanding the impact of altered mitochondrial biology in cancer genomic instability has the potential to make a significant clinical impact and amelioration of current anti-neoplastic therapies.

## Figures and Tables

**Figure 1 cancers-13-01914-f001:**
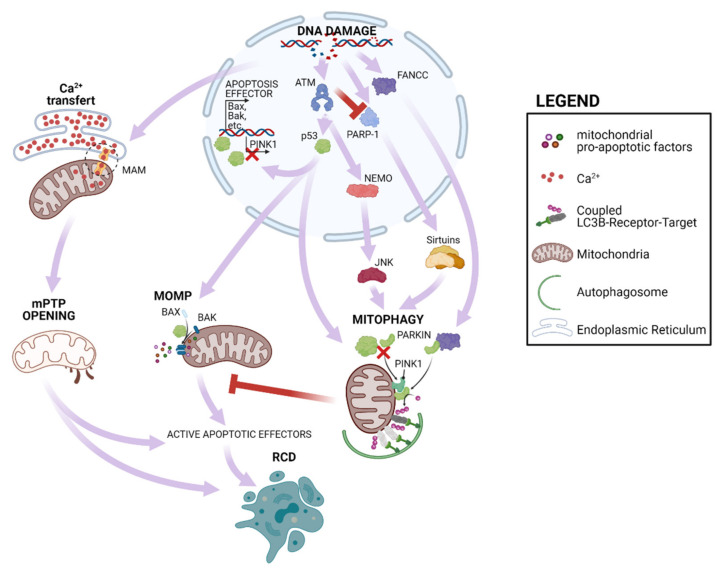
Mitochondrial response to DNA damage. DNA damage can induce RCD by eliciting Ca^2+^ transfer from ER to mitochondria which cause mPTP opening and inducing p53 activation. This favors expression of mitochondrial pro-apoptotic factors as well as MOMP engagement. Further DNA damage response can exert both positive and negative regulation of mitophagy which antagonizes MOMP.

**Figure 2 cancers-13-01914-f002:**
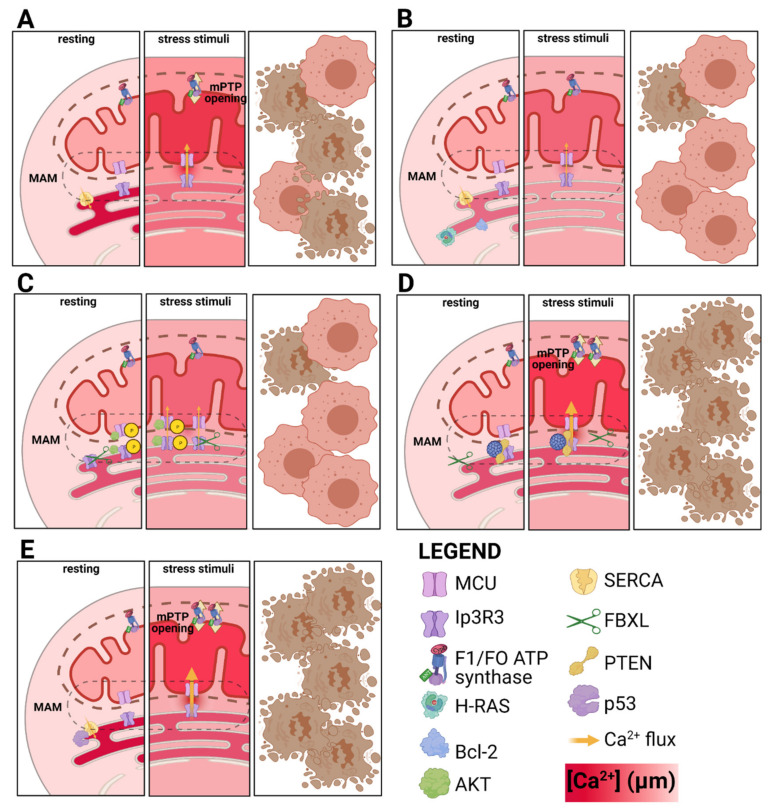
Role of oncogene and tumor suppressor genes in Ca^2+^ dependent RCD. (**A**) In normal cells, stressing stimuli (e.g., UV irradiation) induces the transfer of Ca^2+^ from ER to mitochondria, causing the transition of F1/FO ATP synthase to mPTP, hence the engagement of RCD. (**B**) Elevation of Bcl-2 or H-RAS expression causes the reduction of Ca^2+^ content into ER lumen, causing an impaired Ca^2+^ transfer to mitochondria which fail to properly engage mPTP and RCD. (**C**) AKT activation causes the phosphorylation of Ip3R3 and the MCU member MICU1, making both channels less permeable to Ca^2+^. Similarly, FBXL impairs Ip3R3 stability, and favoring its degradation. Both mechanisms result in a lower Ca^2+^ transfer, while leaving ER Ca^2+^ content unchanged. (**D**) The oncosuppressor genes PTEN and PML counteract the effect of AKT and FBXL, by dephosphorylating and stabilizing Ip3R3. This potentiates the Ca^2+^ accumulation to mitochondria, empowering mPTP opening. (**E**) p53 directly interacts with SERCA, favoring its Ca^2+^ pumping activity within ER lumen. This results in favored Ca^2+^ transfer, mPTP opening and RCD.

**Figure 3 cancers-13-01914-f003:**
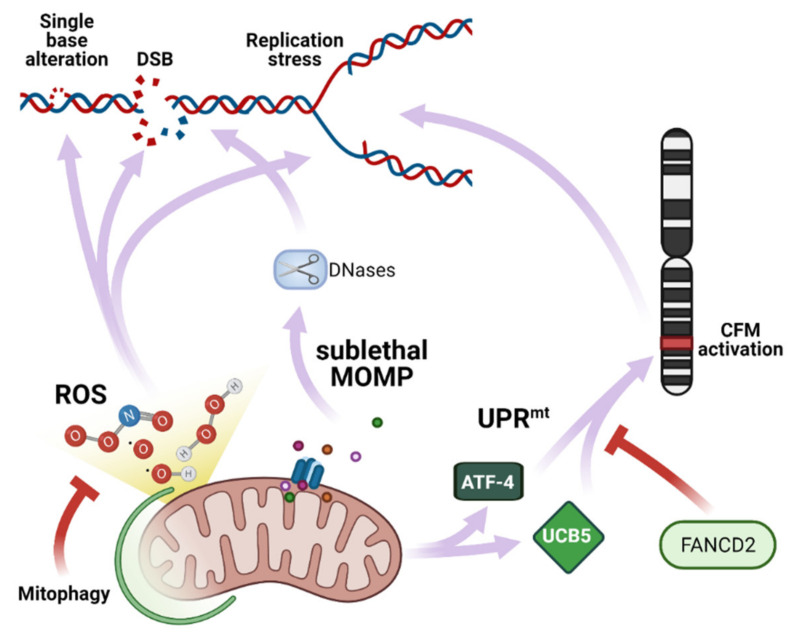
Mitochondrial response to DNA damage. Mitochondria directly induces DNA damage via ROS production, the activation of DNase during sublethal MOMP or activation of CFM mediated by UPR^mt^.

**Figure 4 cancers-13-01914-f004:**
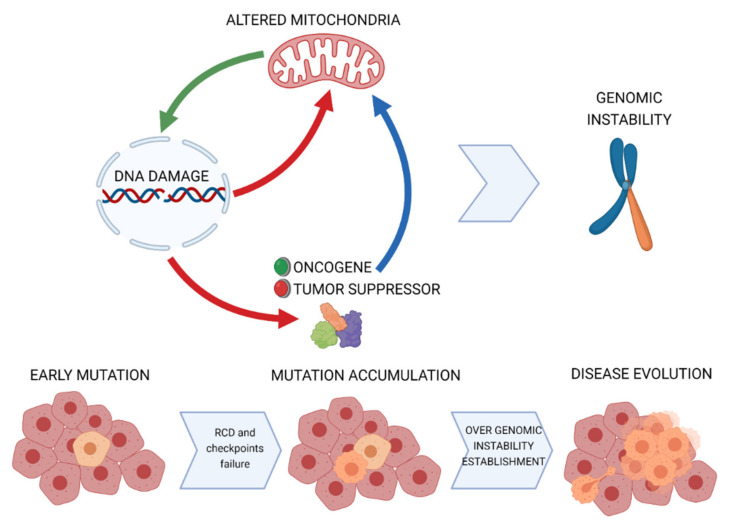
Model for mitochondrial contribution to genomic instability in cancer.

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
