# Peer review of "Mitochondrial Control of Genomic Instability in Cancer"

_cancers, 2021, doi:10.3390/cancers13081914_

Round 1

Reviewer 1 Report

This review summarizes scholarly current knowledge on the impact of DNA damage on mitochondria, and vice versa the role of mitochondria in genomic instability. It is mostly well written except of minor grammatical mistakes and some unclear statements (see below). Figures are well designed and instructive. After appropriate revision this review will certainly serve as an excellent source of knowledge for those interested in the topic.

Critical points

1 the review would benefit from a list of the numerous abbreviations used

2 the role of Ca2+ in mitochondrial physiology, in particual with respect to RCD as described on page 5 seems complex, and description of it was hard to follow. This part might be better appreciated if a schematic figure was added showing the main components and their impact on Ca2+ fluxes.

3 on page 7 the term “near homozygous genome” was introduced. It would be good if authors could describe in one or two sentences what this term precisely means, and explain in more detail why it “could unmask a selective advantage for the metabolic consequences of complex I mutation”.

4 page 8, CSF is introduced here as fragile genomic regions. What is meant by CSF expression or CSF translation? Does this refer to genes present in these regions? Please clarify.

5 Titles of Figure legends are swapped.

Minor

- Frequent mistake: “to occurs” instead of “to occur”, and similar combinations. Please check.

- p. 2 “pathological diseases” replace by “diseases”

- line 106 “armful” should read “harmful”.

- line 170 “can allows” replace by “allows”

Author Response

The authors sincerely thank Reviewer 1 for the useful comments provided. We addressed all at our best and we hope that the editor and reviewer will find, as we do, a significantly improved manuscript.

Will follow a point by point answere to the comments provided.

1 the review would benefit from a list of the numerous abbreviations used

We appreciate the convenient recommendation. A list of abbreviation is now inserted in the manuscript.

2 the role of Ca2+ in mitochondrial physiology, in particual with respect to RCD as described on page 5 seems complex, and description of it was hard to follow. This part might be better appreciated if a schematic figure was added showing the main components and their impact on Ca2+ fluxes.

We apologize for offering a clumpy section. We have now rephrased that paragraph and provided a figure describing the control of oncogene and tumor suppressor genes on Ca2+-dependent RCD (please see the new figure 2).

3 on page 7 the term “near homozygous genome” was introduced. It would be good if authors could describe in one or two sentences what this term precisely means, and explain in more detail why it “could unmask a selective advantage for the metabolic consequences of complex I mutation”.

We have better explained the meaning of “near homozygous genome”. In regards of how it could unmask a selective advantage for complex I mutations, we do not actually have real data to provide a molecular explanations, therefore we rephrased the paragraph providing speculations on what mechanism could lie behind this phenotype.

4 page 8, CSF is introduced here as fragile genomic regions. What is meant by CSF expression or CSF translation? Does this refer to genes present in these regions? Please clarify.

We have now included a brief explanation of the proposed mechanism for CFS breakage. This proposes that transcription of genes at these sites (especially large genes) interfere with the replication machinery causing the stalling of replication fork and formation of secondary structures. Ultimately, this results in a site prone to mechanical rupture or incomplete duplication. Therefore, translation of large genes at CFS is often used as simplified readout of the risk for CFS breakage. We have now improved clarity of this section.

5 Titles of Figure legends are swapped.

We apologize for the mistake, the figure legends are now corrected.

Minor

- Frequent mistake: “to occurs” instead of “to occur”, and similar combinations. Please check.

- p. 2 “pathological diseases” replace by “diseases”

- line 106 “armful” should read “harmful”.

- line 170 “can allows” replace by “allows”

We thank the reviewer for these suggestions, the manuscript grammar have now been carefully checked.

Reviewer 2 Report

As mitochondria is an important organelle playing multiple roles in tumorigenesis, and the interaction between mitochondria biology and genome stability is relatively less studied, an article summarizing such interaction is necessary and interesting to the science community. The current manuscript described the structure and functions of mitochondria, the mitochondrial response to nuclear DNA damage and its consequence, and how mitochondrial ROS causes genomic instability. This manuscript is based on a large number of publications in this field and the figures are clear and informative. However, the reviewer has some minor revision suggestions for this manuscript before it is considered for publication.

  1. When the authors talked about genomic instability in this manuscript, they referred to DNA damage, such as single or double strand breaks. However, some literature also revealed that mitochondria contribute to the epigenetic landscape change during tumor formation. Although this aspect may not be a major focus of this review article, the authors can add a paragraph to describe this point in a concise manner.
  2. The authors did a good job in describing the reciprocal interaction between mitochondria function and genomic DNA damage under physiopathological conditions, mainly from the basic biology point of view. The reviewer suggests adding more description and discussion from a clinical or therapeutic point of view. For instance, using drugs to target mitochondria function, either alone or in combination with other DNA damage-inducing methods, e.g. radiation or chemotherapy. To screen and test mitochondria-targeted drugs for cancer therapy is an evolving topic and many pieces of literature can be referenced.
  3. The references list of this manuscript should be carefully checked as the reviewer believes that there are some missing references, e.g. in line 269-276, the Park et al paper studying U2AF35 is missed, and in line 303-313, a paper studying FANCD2 and CFS is missed for citation.
  4. The manuscript contains some grammar and syntax errors that need to be corrected. For example, in line 170, it should be “mPTP can allow the release” rather than “mPTP can allows the release”.

Author Response

The authors sincerely thank Reviewer 2 for the useful comments provided. We addressed all at our best and we hope that the editor and reviewer will find, as we do, a significantly improved manuscript.

Will follow a point by point answer to the comments provided.

1. When the authors talked about genomic instability in this manuscript, they referred to DNA damage, such as single or double strand breaks. However, some literature also revealed that mitochondria contribute to the epigenetic landscape change during tumor formation. Although this aspect may not be a major focus of this review article, the authors can add a paragraph to describe this point in a concise manner.

This topic is surely of grate interest, we thank the reviewer for pointing it out. While it is reasonable to propose an effect of mitochondria on genomic instability via epigenome regulation there are no study clearly addressing this point. We have therefore described this connection as a possibility to be explored as it might be of significant therapeutical interest.

2. The authors did a good job in describing the reciprocal interaction between mitochondria function and genomic DNA damage under physiopathological conditions, mainly from the basic biology point of view. The reviewer suggests adding more description and discussion from a clinical or therapeutic point of view. For instance, using drugs to target mitochondria function, either alone or in combination with other DNA damage-inducing methods, e.g. radiation or chemotherapy. To screen and test mitochondria-targeted drugs for cancer therapy is an evolving topic and many pieces of literature can be referenced.

This is another significant point and we are glad that Reviewer 2 brought it to our attention. We believe that there is large room for investigation and development for therapeutic intervention in the targeting of mitochondria as adjuvant to DNA damage-inducing method. Still, the consequences of these approaches on genomic instability are still not fully explored (at least to best of our knowledge). We thus expanded the conclusion section discussing this point and its therapeutic potential.

3. The references list of this manuscript should be carefully checked as the reviewer believes that there are some missing references, e.g. in line 269-276, the Park et al paper studying U2AF35 is missed, and in line 303-313, a paper studying FANCD2 and CFS is missed for citation.

We thank the reviewer for underling this mistake, the references have been carefully reviewed.

4. The manuscript contains some grammar and syntax errors that need to be corrected. For example, in line 170, it should be “mPTP can allow the release” rather than “mPTP can allows the release”.

We thank the reviewer for the suggestion, the manuscript have been extensively reviewed also in terms of grammar and syntax.

Reviewer 3 Report

This review critically discusses the role of mitochondrial DNA alterations, mitophagy and DNA damage involved in cancer progression, focusing on the mitochondrial control of genomic instability in cancer.

I have no substantial criticisms to make on the review that in my opinion could be acceptable after a minor revision for few grammatical errors highlighted.

Minor revision

line 8: “Mitochondria are well known to participates

line 9-10: “They indeed can alter the susceptibility of cells to engage regulated cell death, regulates  pro-survival signal transduction pathways and confers metabolic plasticity”

line 12: “….their contribution to the characteristic genomic instability that underlies the evolution”

line 108: “Apoptosis is an ultimate cellular decision, which consists in the physical loss

In lines 158-160 is written: "Those three enzymes represent rate-limiting steps thus controlling the feeding of electrons into the respiratory chain and the generation of the proton gradient across the inner membrane, in turn necessary for Ca2+ uptake and ATP production

What are those three enzymes? It is appropriate to specify which are the three enzymes not mentioned and I suggest to add a reference on the subject at the end of the sentence.

lines 181-183: “On opposite, Bcl-2 overexpression was demonstrated to favors Ca2+ leak from ER, dampening the execution of apoptosis [54]. Again, H-RAS was proposed to impairs tumor transformation

lines 323-324: “Especially, the cancer cells select for impaired mitochondrial engagement of RCD in response to DNA damage and exacerbates mitochondrial phenotypes”

Note: the "s" to be removed are highlighted in yellow and the "s" to be added are highlighted in green

Author Response

The authors sincerely thank Reviewer 3 for the useful comments provided. We apologize for having offered a manuscript with many mistake in the first place. We performed extensive review of the whole manuscript and Ca2+ sensitive enzymes of the TCA cycle properly mentioned. We hope that the editor and reviewer will find, as we do, a significantly improved manuscript.

Round 2

Reviewer 1 Report

The revised version addressed critical points raised bythis reviewer and is now acceptable for publication.